# Trends in the costs of drugs launched in the UK between 1981 and 2015: an analysis of the launch price of drugs in five disease areas

Derek J Ward,[1] Lucy Doos,[2] Andrew Stevens[1]

¹Institute of Applied Health Research, University of Birmingham, Birmingham, UK
²College of Medical and Dental Sciences, University of Birmingham, Birmingham, UK

**Correspondence to**
Dr Derek J Ward;
d.j.ward@bham.ac.uk

## ABSTRACT

**Objectives** To investigate the trend in the launch price of new drugs for five common health conditions.

**Design** Cross-sectional study using data on new drugs launched in the UK between 1981 and 2015 for hypertension, asthma, rheumatoid arthritis, schizophrenia and colorectal cancer.

**Data and sources** All drugs marketed in the UK between 1981 and 2015 (inclusive), and licensed specifically for the treatment of one of the five chosen conditions were included in the study. Newly launched medicines and their launch prices were identified by hand-searching all editions of the British National Formulary in addition to searching the websites of relevant regulatory agencies (European Medicines Agency and Medicines and Healthcare products Regulatory Agency). The launch price in UK pounds for a 28-day supply of each medicine at a typical or usual maintenance dose was adjusted for the effects of general inflation using the gross domestic product deflator series.

**Results** 104 drugs were included in our study with a mean inflation-adjusted 28-day launch price of £288 (SD £678). The launch price of new drugs varied significantly across the five conditions, with drugs for hypertension having the lowest mean price (£27) and drugs for colorectal cancer having the highest mean price (£1590) (p<0.001). There were large increases in launch prices across the study period, but the magnitude and pattern was markedly different between therapeutic areas. Biological drugs represented 13.5% of all included drugs and had a significantly higher launch price than non-biological drugs (£1233 vs £141, p<0.001). 22.1% of included drugs were first-of-kind and had a significantly higher launch price than follow-on drugs (£768 vs £151) (p<0.0001).

**Conclusion** Drugs prices continue to increase across different therapeutic areas. This has some association with novelty, but, it is not clear if this increase in price is associated with medical benefits.

## Strengths and limitations of this study

► The timeline of this study enables a very long-term view of drug pricing that goes beyond previously published work.
► This study used the British National Formulary to identify new drugs and new licensed indications for existing marketed drugs and is therefore likely to represent a comprehensive view of drug pricing in the UK.
► This study is restricted to publicly available pricing data in the UK; the actual price paid by healthcare providers for drugs may vary from this and the results may not be applicable to other settings.
► This study chose to focus on five health conditions in a pragmatic way, and the results may not be generalisable to drugs licensed for use in other health conditions.

## BACKGROUND

Over last few decades, the expense on healthcare has risen faster than economic growth in many developed countries.[1 2] Internationally, expense on pharmaceuticals represents a significant proportion of the total healthcare budget.[3] For example, high-income countries within the Organisation for Economic Co-operation and Development spend, on average, 18% of their total healthcare expenditure on medicines and this figure can reach up to 80% in some low- and middle-income countries.[3] In the UK, the expenditure on medicines represented 11.6% of total healthcare expenditure in 2008.[1]

Worldwide, affordability is a major component of ensuring access to essential medicines for many conditions.[4 5] Affordability reflects both price and volume, and many publicly funded healthcare providers, including the UK National Health Service (NHS), aim to provide effective treatment at a price that represents value for money.[6] Healthcare systems in many countries, including in the UK, use a variety of cost-saving and cost-containing measures in order to counter financial challenges. A government-wide agreement with industry to cap increases in overall expenditure on branded medicines (the Pharmaceutical Price Regulation Scheme) and ensuring approval and reimbursement

BMJ

of drugs is dependent on an assessment of clinical and cost-effectiveness using Health Technology Assessment (HTA), which may include restrictions on patient eligibility.[6] An understanding of the drivers of medicine prices is therefore important, particularly when countries and policy makers are seeking to develop pricing policies that improve both the availability and the affordability of such medicines.[3]

The increasing cost of pharmaceuticals used to manage a number of common conditions has received increasing attention in recent years.[4 7] In the USA, retail prescription drug spending accelerated in 2014, growing 13.1% in 1 year, representing the largest annual increase since 2003.[8] According to a recent report that considered the impact of changes in the pharmaceutical industry and its impact on healthcare payers in the USA,[8] this increase was the result of increasing demand and changes in patient behaviour, both of which had a significant impact on drug expenditures.

However, innovation or novelty is also a factor, as pricing of first-of-kind drugs was noted to be one of the most important factors driving this trend.[8] Studies on rising overall pharmaceutical expenditure seen during the 1990s to mid-2000s in North America and Europe have highlighted the role played by both increased utilisation and the adoption of newer, more expensive medicines.[9 10] The high cost of newer agents has been identified by some commentators as the key component of rising per-patient pharmaceutical costs for cancer chemotherapy and the treatment of diabetes mellitus, glaucoma, psychosis, multiple sclerosis and haemophilia.[11 12] However, other studies suggest increasing utilisation as the major driver of rising costs in the treatment of hypertension, hyperlipidaemia, depression and rheumatoid arthritis, where increasing overall pharmaceutical spending has not necessarily been accompanied by increased per-patient spending.[13–15]

Increases in the cost of newer medicines has been variously explained as due to increasing regulatory and technology assessment requirements, research and development costs, attrition rates, production costs, the patent system and marketing practice.[16–22] To further investigate trends in the price of pharmaceuticals, we sought to describe long-term, 35-year trends in the launch price for all new medicines marketed in the UK to treat one of five common conditions.

## METHODS
### Selection of disease areas and identification of new drugs
All medicines marketed in the UK for the first time between 1981 and 2015 (inclusive) for the treatment of one the following five conditions were included in the study using International Statistical Classification of Diseases version 10 (ICD-10)[23]: (1) asthma (ICD-10 code J45), (2) colorectal cancer (ICD-10 code C18), (3) essential hypertension (ICD-10 code I10), (4) rheumatoid arthritis (seropositive rheumatoid arthritis (ICD-10

codes M05 and M06)) and (5) schizophrenia (ICD-10 code F20). These diseases were selected by consensus of the six members of the National Institute for Health Research Horizon Scanning Research and Intelligence Centre research group to represent a range of physical and mental health disorders for which significant pharmaceutical innovation over the period of interest was known to have occurred.

Our study included medicines specifically licensed for the treatment of one of these conditions at the time of first UK launch (as described in the Summary of Product Characteristics (SmPC) published by the European Medicines Agency (EMA)[24] and Medicines and Healthcare products Regulatory Agency (MHRA),[25] as appropriate as well as existing licensed drugs that subsequently received marketing authorisation for one of the five study conditions, including those that have subsequently been withdrawn.

Medicines were identified by hand-searching all editions of the British National Formulary (BNF)[26] published between 1981 and 2015. The BNF lists all medicines available for prescription and dispensing within the UK irrespective of whether they are available on the NHS; the date of first appearance in the BNF (for new medicines) or the date where the indication was amended in the BNF (for existing licensed medicines subsequently receiving marketing authorisation for one of the specified conditions) was taken to be the year of launch in the UK. Medicines listed in BNF edition 1 were considered to be already be available for use in 1981, and therefore excluded from the study. This approach to identifying all relevant new medicines was supplemented by searches of the electronic Medicines Compendium (Datapharm Communications),[27] commercial pharmaceutical databases (Adis Insight (Springer))[18] and Pharmaprojects (Informa Healthcare),[29] the websites of relevant regulatory agencies (EMA and MHRA),[24 25] and searches for clinical practice guidelines issued by the National Institute for Health and Clinical Excellence (NICE)[30] and relevant specialist clinical societies and associations.

We had no patient or public involvement as part of this study.

### Price calculation
To allow price comparisons within each disease area, the launch price in UK pounds for a 28-day supply of each medicine at a typical or usual maintenance dose (including cycles or courses where relevant) was calculated from the unit price provided in the edition of the BNF in which the medicine first appeared. Initial loading or lower introductory doses were ignored. The published SmPC and/or relevant clinical guidelines were consulted when no usual dose was listed in the BNF. The dose range mid-point was used when no usual dose could be established (adjusted to be achievable using available dose formulations). In all cases, the price was calculated using the least expensive combination of available dose

formulations (where relevant, this assumed wastage, a body weight of 76.9 kg, and body surface area of $1.7\,m^2$).

The 28-day launch prices for all medicines were adjusted for the effects of general inflation using the gross domestic product (GDP) deflator series.[31] GDP deflators serve as a measure of inflation within the economy as a whole for a given time period and were used to adjust prices to 2015 values, thereby allowing direct comparisons between prices at different time points.

We excluded new combinations of existing licensed medicines and those not found in the BNF.

### Data handling and analysis

Data on drug name, whether it was newly launched or an existing licensed and available product, the mechanism of action or drug class, whether the drug was a biological agent, launch year, launch price for a typical 28-day supply and 2015-adjusted 28-day launch price were extracted and entered onto an Excel spreadsheet. Where the mechanism of action or drug class was not stated in the BNF, it was obtained from the commercial Pharmaprojects database. Statistical analysis was carried out using IBM SPSS V.21 for Windows. Trends in adjusted 28-day launch prices were initially explored using simple descriptive statistics and scatter plots. Descriptive analyses were presented as means and SD for normally distributed continuous variables. Significant differences were determined using analysis of variance for continuous normally distributed data and $X^2$ for dichotomous variables.

### RESULTS

One hundred and four drugs met our inclusion criteria, representing approximately 10% of all new drugs launched for all conditions over the period of the study.[32] They were launched with an inflation-adjusted 28-day price that ranged from £2.20 (methotrexate for rheumatoid arthritis, 1992) to £4200.70 (bevacizumab for colorectal cancer, 2006) (table 1). Nearly one third of the drugs were launched for hypertension (34.6%), while just 10.6% were launched for colorectal cancer. There were statistically significant differences in the mean inflation-adjusted 28-day launch price by condition, with drugs for hypertension having the lowest mean price (£27) and drugs for colorectal cancer having the highest mean price (£1,590) (p<0.0001).

### Drug characteristics

The majority of the drugs included in the study were newly launched (92.3%) as opposed to new licensed indications for existing marketed drugs (9.6%). The mean inflation-adjusted 28-day price for newly launched drugs was higher than that for new licensed indications for existing marketed drugs. This pattern was observed across all three relevant conditions (asthma, rheumatoid arthritis and colorectal cancer), but the differences were not statistically significant (table 2).

Fourteen (13.5%) of the included drugs were biological agents, almost two thirds of which were for rheumatoid arthritis, and none of which were for hypertension or schizophrenia. Over the three relevant conditions, biological drugs had a significantly higher mean inflation-adjusted 28-day launch price than non-biological drugs (£1233 vs £141, p<0.001).

Newly launched biologics had nearly twice the mean inflation-adjusted 28-day launch price of the two newly licensed indications for existing marketed biological drugs (£1322 vs £691), however this difference was not statistically significant.

### Pattern of change in pricing

A large increase in inflation-adjusted 28-day launch prices was observed when all drugs included in this study were considered together. However, the magnitude and pattern of change varied considerably between the five therapeutic areas (figure 1).

For hypertension, following the introduction of captopril in 1981, the typical price of newly launched drugs generally fell (with some variation) until the early 1990s, after which prices appeared to stabilise. For asthma, there was a single high-cost outlier (omalizumab, launched in 2006), but there was also a sharp increase in launch price in 1987 with the launch of nedocromil sodium (a cromoglicate), which was nearly three times the inflation-adjusted price of the previously launched drug. The overall trend in the price of newly launched drugs for schizophrenia was broadly flat. However, there were some significant variations across the period of interest.

**Table 1** Number of drugs and their 28-day inflation-adjusted launch price by selected condition launched in the period 1981 to 2015 (inclusive), includes newly licensed drugs as well as new licensed indications for existing marketed drugs. Prices adjusted to 2015 values using the gross domestic product deflator series.

| Selected condition | n (%) | Mean price (SD) | Median price (range) |
|---|---|---|---|
| Hypertension | 36 (34.6) | £27.01 (13.09) | £23.48 (7.40–59.10) |
| Asthma | 18 (17.3) | £64.73 (175.10) | £21.32 (8.60–764.50) |
| Schizophrenia | 15 (14.4) | £130.31 (86.56) | £131.25 (19.40–331.80) |
| Rheumatoid arthritis | 24 (23.1) | £347.16 (414.24) | £69.71 (2.20–1052.20) |
| Colorectal cancer | 11 (10.6) | £1590.30 (1426.65) | £1105.20 (83.04–4200.70) |
| Total | 104 | £287.67 (678.34) | £37.76 (2.20–4200.70) |

**Table 2** Mean 28-day inflation-adjusted launch price by selected condition according to biological vs non-biological drug type, first-of-kind vs follow-on drug, and new drug vs new indication for existing marketed drug, 1981–2015.

| Selected condition | Biological drug | Non-biological drug | First-of-kind | Follow-on | New drug | New licensed indication |
|---|---|---|---|---|---|---|
| Hypertension | | | £31.50 | £26.50 | | |
| | | | P=0.47 | | | |
| Asthma | £764.50 | £23.60 | £283.90 | £20.90 | £70.40 | £19.20 |
| | P<0.001 | | P=0.01 | | P=0.71 | |
| Schizophrenia | | | £331.80 | £115.90 | | |
| | | | P=0.01 | | | |
| Rheumatoid arthritis | £813.90 | £211.90 | £451.80 | £227.40 | £389.70 | £329.90 |
| | P=0.001 | | P=0.31 | | P=0.79 | |
| Colorectal cancer | £2225.48 | £1227.35 | £2368.44 | £941.86 | £1741.03 | £83.04 |
| | P=0.28 | | P=0.10 | | P=0.29 | |
| All conditions | £1232.76 | £140.65 | £768.25 | £151.21 | £293.19 | £221.34 |
| | P<0.001 | | P<0.001 | | P=0.78 | |

In particular, two drugs launched in 1990 showed a great discrepancy in inflation-adjusted 28-day launch price, with the price of clozapine (£332.80, the first atypical antipsychotic launched) being nearly 10-fold higher than that of loxapine (£31.75), which was launched the same year. For rheumatoid arthritis, launch prices experienced a marked upward step in 1999 with the arrival of infliximab (tumor necrosis factor [TNF]-α inhibitor), the first biological agent licensed for this condition. This was followed by a slight decrease in the inflation-adjusted

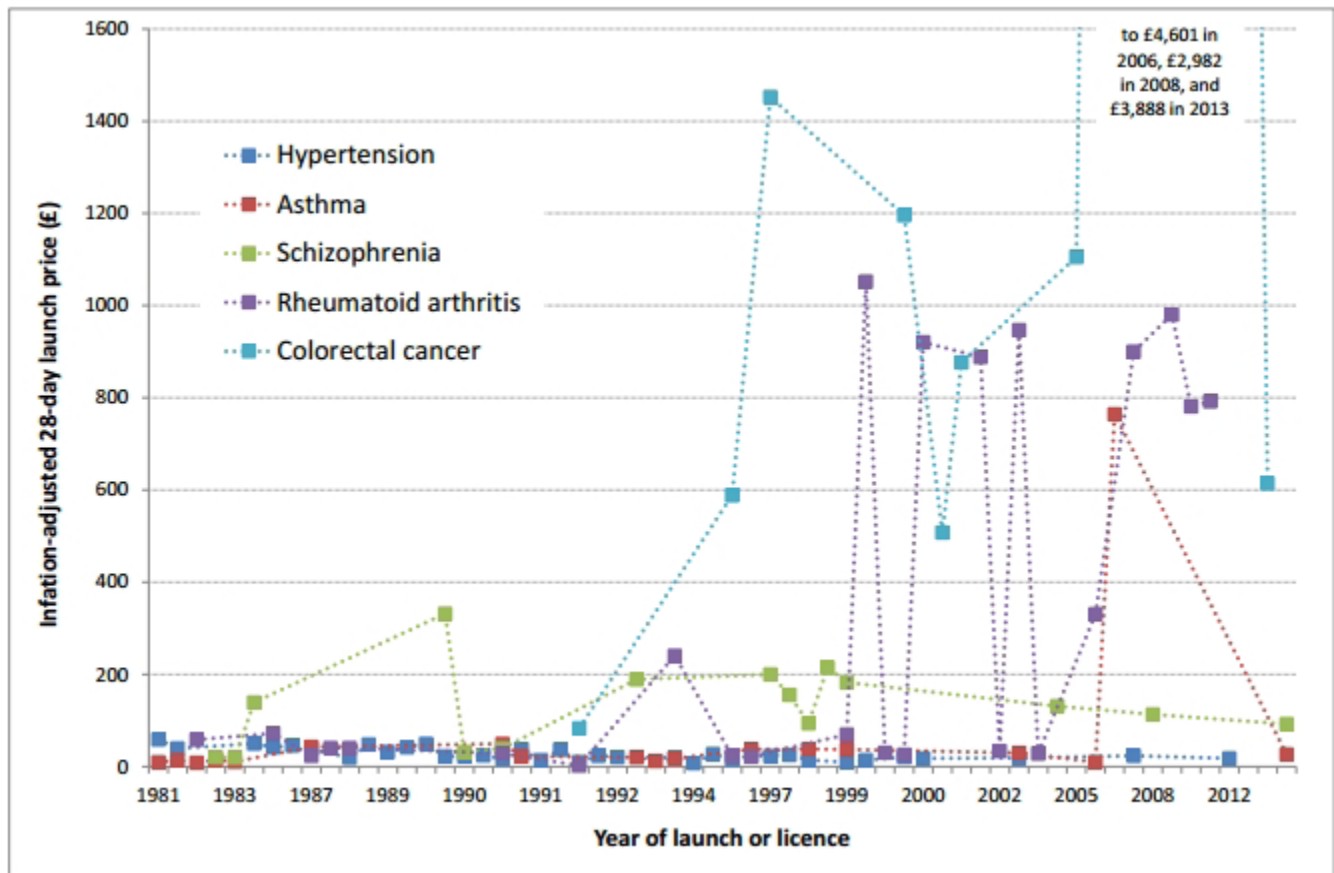

**Figure 1** Inflation-adjusted 28-day launch price of newly licensed drugs and new licensed indications for existing marketed drugs for one of five selected conditions, 1981 to 2015 (inclusive).

price the following year when the next drug was launched (by 12.5%). Launch prices for drugs licensed to treat colorectal cancer rose increasingly rapidly in the two decades following the mid-1990s. In particular, the introduction of bevacizumab in 2006, the second biological drug for this condition, resulted in a nearly threefold increase in the inflation-adjusted 28-day launch price compared with drugs launched in the previous few years.

### Drug class/mechanism of action

The 104 drugs included in our study represented 34 different drug classes or mechanisms of action, 17 of which were represented only once. There were significant variations in the inflation-adjusted 28-day launch price for drugs by drug class/mechanism of action; these differences were statistically significant across all but one of the therapeutic areas (the exception being drugs for colorectal cancer) (supplementary appendix 1).

Among the 36 drugs from nine drug classes included for hypertension, α1-adrenergic receptor antagonists had the highest mean inflation-adjusted 28-day launch price (£36.50) followed by angiotensin converting enzyme (ACE) inhibitors (£35.00). For asthma, the anti-IgE humanised monoclonal antibody, omalizumab, had the highest inflation-adjusted launch price by far (£764.45). The next most expensive was the single cromoglicate (£42.74), followed by leukotriene receptor antagonists (£37.58) and selective β2-agonists—long acting (£31.21). The lowest mean inflation-adjusted 28-day launch price was for selective β2-agonists—short acting (£11.62). For rheumatoid arthritis, nearly one third of drugs (29.2%) were non-steroidal anti-inflammatory drugs , and there was a marked downward trend in their launch prices between 1985 and 1996 (mean adjusted 28-day launch price for the whole period £40.76). TNF-α inhibitors were the second common group of drugs launched for rheumatoid arthritis, and these had a mean inflation-adjusted launch price of £898.53, though single examples of an interleukin (IL)-1 inhibitor, an inhibitor of T-cell co-stimulation and IL-6 inhibitors were priced even higher. Almost one third of all new drugs launched for colorectal cancer were thymidylate synthase inhibitors, which had a mean inflation-adjusted 28-day launch price £513.78. The launch price for this class of drugs showed a dramatic, more than 10-fold increase between 1992 and 2001, but even so the mean launch price was still the lowest of all drug classes/mechanisms of action identified for this condition. Drugs for schizophrenia were grouped into two broad categories, recognising the difficulty of ascribing a specific class or individual mechanism of action. Atypical antipsychotics had significantly higher inflation-adjusted 28-day launch price (£156.93) than first generation antipsychotics (£23.80) (p<0.01).

### First-of-kind and follow-on drugs

Less than a quarter of the drugs included in our study (22.1%) represented the first of a new class of drugs or a new mechanism of action (first-of-kind). However, these drugs had a significantly higher inflation-adjusted 28-day launch price than follow-on drugs (£768 vs £151, p<0.0001) (table 2). However, within therapy areas, a statistically significant difference between first-of-kind and follow-on drugs was only observed for asthma and schizophrenia (both p<0.01). First-of-kind drugs represented almost half of all newly launched drugs for colorectal cancer, a proportion that was greater than for any other indication.

We identified nine examples where the first-of-kind and at least one follow-on drug were launched in the time period of this study; two each of these drug classes/mechanisms of action were licensed for colorectal cancer, asthma, hypertension and rheumatoid arthritis, and one was licensed for schizophrenia. In general, it was apparent that inflation-adjusted launch prices generally fell over time, particularly as the time from when the first-of-kind was launched increased (figure 2). The overall decrease in inflation-adjusted 28-day launch price from the first-of-kind to the final launch of a follow-on varied between -£2.00 (selective cyclo-oxygenase-2 inhibitors for rheumatoid arthritis) to -£259.99 (TNF-α inhibitors for rheumatoid arthritis), and in relative terms varied between −6.6% (selective cyclo-oxygenase-2 inhibitors for rheumatoid arthritis) to −83.9% (ACE inhibitors for hypertension). The exceptions were the two epidermal growth factor receptor (EGFR) inhibitors for colorectal cancer, where panitumumab was priced over 160% higher than cetuximab, the first-of-kind.

## DISCUSSION
### Main findings

Our findings show that while launch prices of drugs have generally increased over the study period, after controlling for inflation there were marked differences according to therapeutic area, with some showing no real-term increase. In addition, the launch price of first-of-kind drugs is consistently higher than that of other drugs, and that biological agents are considerably more expensive at launch that other drugs marketed for the same condition.

Pharmaceutical companies argue that the price of new drugs is set at a level needed to recoup the vast costs incurred in bringing that drug to market, as well as fund future research and development.[33–35] These costs have risen over time for a number of reasons, including the increasing complexity and length of clinical trial programmes to meet the needs of regulators and HTA bodies,[36–39] so that, by some estimates, the rate of new drug launches per R&D spend has fallen by around a half every 9 years since the 1950s.[40] However, R&D costs per drug candidate are not transparent for commercial reasons, and the oft quoted cost of bringing a new drug to market may fail to properly account for publicly funded basic and translational research.[34] In our study, the mean inflation-adjusted 28-day launch price for drugs indicated for colorectal cancer was almost 59 times greater than the price of those for hypertension. We cannot directly

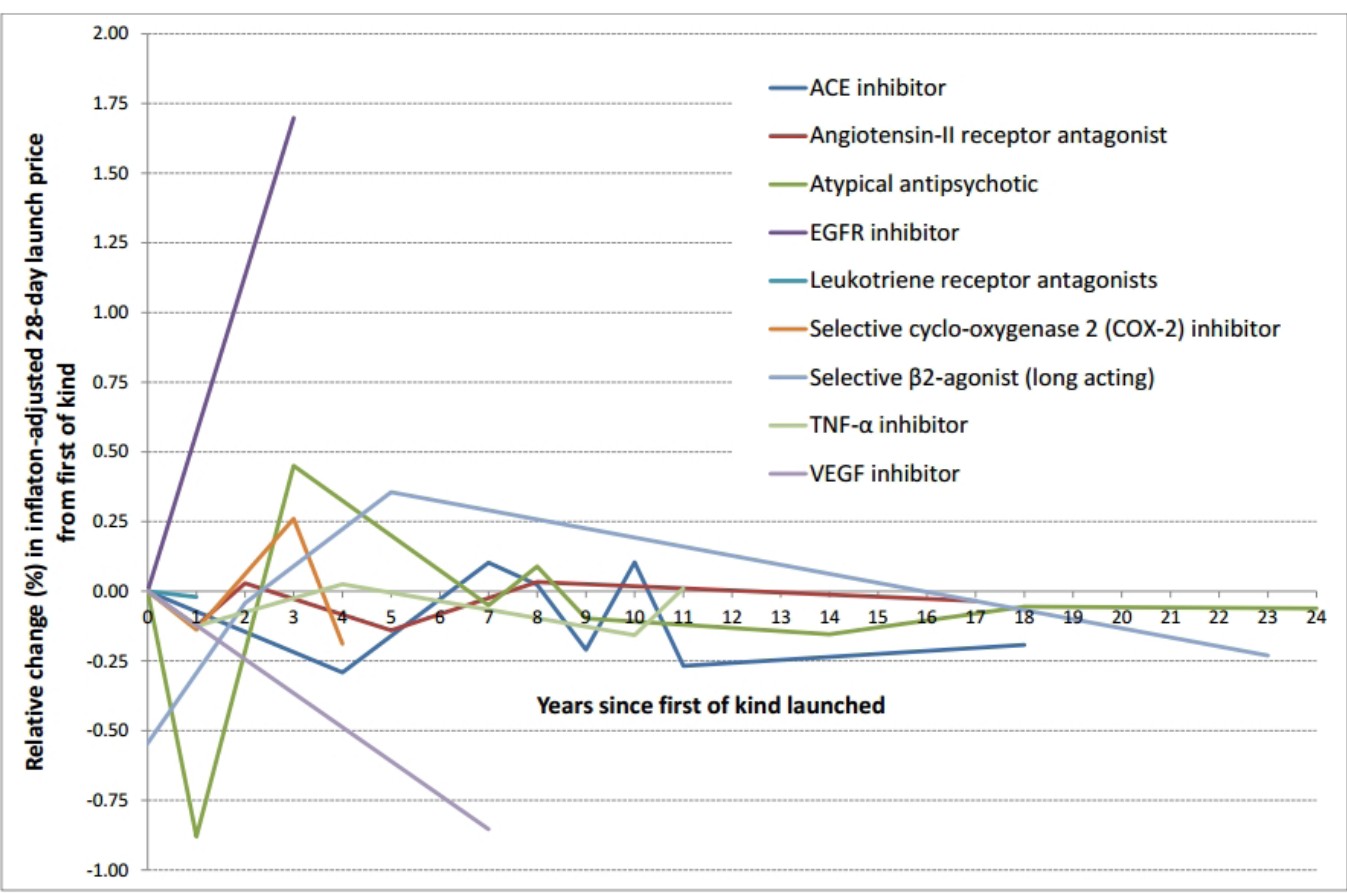

**Figure 2** Relative (%) change in inflation-adjusted 28-day launch price of follow-on drugs compared with first-of-kind drugs by number of years since launch of first-of-kind. ACE, angiotensin converting enzyme; EGFR, epidermal growth factor receptor; TNFa, tumor necrosis factor; VEGF, vascular endothelial growth factor.

account for different drug development costs across different disease areas, but this finding may reflect the headroom for therapeutic gain in colorectal cancer when compared with hypertension. Our data appears to agree with the general observation made by commentators that managing cancer is expensive and the prices for drugs associated with cancer represent a great burden to healthcare systems.[36 41] Howard *et al*[36] reported that the average adjusted launch price of anticancer drugs increased by 10% annually between 1995 and 2013, which is considerably less than the increase in inflation-adjusted launch price of drugs for colorectal cancer we observed in the period 1985 to 2013. Launch prices spiked in 2006 with the introduction of the biological drug, bevacizumab, which might be expected given the increased development and manufacturing costs associated with biological agents.[34] This pattern of increased drug costs associated with the introduction of the first biological drug mirrored that seen in rheumatoid arthritis, but unlike rheumatoid arthritis, commentators have argued that this pressure on prices seen in cancer is not related to the magnitude of expected health benefit.[36]

We also found that first-of-kind drugs were more expensive at launch and that where our data included both first-of-kind and follow-on drugs, the price of subsequent

launches was lower but still rather closely related to that of the first launch. This pattern is readily explained by the market into which new and follow-on drugs are launched, with first-of-kind drugs competing with established drugs for the same condition or creating a new market if no equivalent treatment exists. The premium given to first-of-kind drugs could be justified if payers are willing to pay more to receive a substantial therapeutic advance compared with existing drugs or other treatment options.[33] In our data, this may partly explain the very marked differences between areas where there is limited scope for new drugs to markedly improve care. The idea of 'rewardable innovation' suggests that health services and patients place value on improvements in a wide range of domains, including those related to new molecular structures, drug synthesis, pharmacodynamics, pharmacokinetics, drug delivery, pharmacogenetics and application in clinical practice.[42] However, many newly launched drugs do not offer any therapeutic advance for patients[41] and even many first-of-kind drugs do not provide any clinically significant innovation.[43]

### Strengths and limitations of the study
We believe our study includes all drugs newly launched in the UK for the five indicated conditions over a 35-year

period. However, we chose the five conditions on a pragmatic basis, and generalisability from our findings cannot be assumed, particularly in drugs for rare or orphan conditions, which were not part of our study. In addition, the published list price for drugs may not be the actual price paid by NHS organisations, particularly for the more recently launched drugs, as this fails to account for the effects of local price negotiations and the increasing use of patient access schemes. These predominate in new cancer drugs, and tend to be of an order sufficient to yield a cost-utility ratio acceptable to NICE, but their impact would tend to blunt the apparent increase in price over time for a particular condition, whereas our data still show a marked increase in the inflation-adjusted price of new drugs for colorectal cancer. Finally, we acknowledge that our decision to standardise comparisons on the basis of a 28-day supply of medicines could be criticised as failing to note that treatment for some conditions and with some medications could potentially be lifelong, and hence generate much higher total costs for payers. Indeed, some innovations in cancer could be life-extending, and so overall costs could escalate further, although with important health gains. However, our principal comparisons were within disease areas, and in general, this means treatments are given for a similar duration or course.

## Conclusions

While prices for new drugs continue to increase overall, this masks significant disparities between therapeutic areas. It is unclear to what extent these patterns represent developers passing on real differences in development and manufacturing costs or different pricing strategies based on an assessment of the current market and payers' willingness to pay more for some conditions. Increased prices could be justified where new drugs meet an unmet clinical need[42 43] and could encourage developers to focus efforts in priority fields, where higher risk might reasonably be rewarded by higher returns. Publicly funded healthcare systems demand value for money, but existing pricing systems lack the transparency to ensure access to drugs at reasonable and justified prices.

**Contributors** DJW and AS conceived the original study idea, and LD contributed to the development of the study design and methods. LD extracted the data, which was independently reviewed by DJW. LD and DJW conducted the analyses. All authors were involved in the interpretation of the results. LD produced the initial draft of the paper, which was then circulated repeatedly to all authors for critical revision. LD, DJW and AS read and approved the final version. All authors had full access to all of the study data (including statistical reports and tables), and can take responsibility for the integrity of the data and the accuracy of the analysis. DJW is the guarantor.

**Funding** The study was undertaken as part of the research programme of the National Institute for Health Research Horizon Scanning Research & Intelligence Centre (NIHRHSRIC). HSRIC is funded by the NIHR The NIHR had no role in the study design; in the collection,analysis and interpretation of the data; in the preparation of themanuscript; or in the decision to submit the article for publication.This article presents independent research funded by the NIHR. The views expressed are those of the authors and not necessarily those of the National Health Service, the NIHR or the Department of Health.

**Competing interests** All authors have completed the ICMJE uniform disclosure format and declare: all authors have financial support from the NIHR for the

submitted work; all authors have no financial relationships with any organisations thatmight have an interest in the submitted work in the previous 3 years; and all authors have no other relationships or activities that could appear to have influenced the submitted work.

**Patient consent for publication** Not required.

**Provenance and peer review** Not commissioned; externally peer reviewed.

**Data sharing statement** All data are included in the supplementary file.

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
