## [Reviewer comments · BMJ Open]

ARTICLE DETAILS

TITLE (PROVISIONAL)	Trends in the costs of drugs launched in the United Kingdom between 1981 and 2015: an analysis of the launch price of drugs in five disease areas
AUTHORS	Ward, Derek; Doos, Lucy; Stevens, Andrew

VERSION 1 - REVIEW

REVIEWER	Dr Philip Savage BSUH UK
REVIEW RETURNED	19-Nov-2018

GENERAL COMMENTS	Thank you for asking me to review this paper. The major increase in the numbers of new drugs and the relative escalation in costs over the past 10-20 years are of significant clinical and economic importance Adding data and discussion to this debate is important and the authors have aimed to look at the wider fields of a number of medical specialties and allow comparisons in development and economics between them. I would support this aim but feel a number of changes to the paper could be made to improve the impact 1/ Background information It would be helpful to explain the choice of diseases used in the study. How many patients are on treatment for these conditions in the UK and how long does an average course of therapy last. 2/ Study data The title outlines that we are looking at New Drugs, however a number of the drugs in the analysis are old drugs with new indications (ie Methotrexate an old and cheap drug with an added indication for RA vs Rituximab a new and expensive drug with an added indication for RA). The costs of these drugs were already determined previously and are no different for the new indications. Should these drugs with new indications rather than new drugs be included in the study? The database may also include one error. 5-FU has been around a long time and I think may have been in the BNF since the 1970s. It would also be good to know the total number of new drugs for all indications licenced during the study period 3/ Cost for 28 days Whilst this is important, I am not sure if this is the only important comparison for economics. Some of the conditions (hypertension) are associated with very long treatment periods (decades) whilst
--

	colorectal cancer is usually only treated for a number of months. Some comment on the overall cost of treatment with a new therapy would also be informative I may need to recheck my data but I think the BNF price for a 28 day cycle (21 day treatment) of Regorafenib was £3744 at launch but is £1296 in the data set. Additional info To help get a wider of view of this, it might be helpful to consider adding some more detail to the discussion The authors describe the costs of developing a new drug and this impact but do not supply an estimates of this cost. This would aid the discussion UK prices. It may be worth adding in some discussion on how UK drug prices are set. The UK has a number of price control mechanisms PPRS and NICE and some readers may be unaware of the very signifcant impact these have on drug prices. Overall this comparison showing the impact of new technology and new drugs on the economics of differing types of illness is welcome and should be of value to those looking at the overall health economy, rather than only from an individual specialist viewpoint
--	---

REVIEWER	Jan Norum UiT - The Arctic University of Norway Tromsø, Norway
REVIEW RETURNED	17-Dec-2018

GENERAL COMMENTS	The title indicates a much larger study. It does not indicate that only five health conditions were focused. Many readers, looking at the heading, may be excited, but disappointed when reading the paper. The title should be rewritten. The major weeknes of this work is that the price paid by the National Health Serice (NHS) is not known. Today negotiation over price is common in most countries and secret pricing is a hugh problem when health economic analyses are undertaken. The price actually paid may be less than half of the official list price. I strongly recommend that this weakness is more elucidated and the price level actually paid should be given or at least indicated. The secret pricing and the fact that the results of the negotiations over price are forbidden to publish, causes a signiicant problem to health administrators and economists struggling with the documentaion of cost effectiveness in Health Technology Analyses (HTAs). The great challenge is not the rising costs/prices, but the fact that almost no-one pay these prices. The 28 days perspective is very short. Some therapies are administered for a limited time frame (in ex. colorectal cancer therapy) and others are administered for decades (hypertension, arthritis etc.). Despite a low cost per month, the cost to the society may be high. This should also be more strongly pointed at.
---

REVIEWER	Alessandra Ferrario Harvard Medical School, USA
REVIEW RETURNED	19-Dec-2018

GENERAL COMMENTS	This is an interesting and well executed study. I think the study could have added more value if it had collected also some indicators of the drug added clinical value to put prices into perspective and if it has evaluated price development for the individual drugs over time. So overall the study provides interesting longitudinal evidence on launch prices and raises important questions as to price drivers, it does less in terms of providing answers in regards to determinants launch prices over time (not the objective of the study but an important question to answer from a policy and practice perspective). Background p.5 Can you add the relevant references to the last two sentences: "In the USA, [..]" "According to a recent [..]" It is important to distinguish clearly between price and expenditure. The focus of the article is on the price, price can be a driver or increased expenditure but expenditure could also increase independently of a price increase (move towards newer and more expensive products). I think it would be good to clearly distinguish price from expenditure as well as their respective drivers. I would perhaps avoid to introduce a third term 'cost' to avoid confusion. In the second paragraph you talk about affordability. Affordability is driven by price and volume however the rest of the paragraph focuses on price only. There are other tools available to health care systems to ensure access (of which affordability is a major component as you say). Use of limited formularies (essential medicine lists) is a tool to limit use to the most cost-effective medicines which contributes to better affordability for health care system for example. I suggest to improve the flow of the paragraphs leading up to the objective of the study: expenditure on medicines is increasing, there are several drivers, price is one of them, your article focuses on prices of new medicines (you may also want to acknowledge that there have been price increases for off-patent medicines but that this is not the focus of your study). It may not always be possible, but where these are available, I would suggest to use more recent references in your background (particularly when referencing papers on expenditure and price trends). Here some examples of recent articles you may want to consider citing: http://care.diabetesjournals.org/content/early/2018/05/03/dci18-0019 https://www.ncbi.nlm.nih.gov/pubmed/29016226 https://www.ncbi.nlm.nih.gov/pubmed/28605615 Methods Overall, methods are clear and well described. p.8 'launch price for a typical 28-day supply and 2015 adjusted 28-day launch price were extracted' I am not sure if I understand this correctly: You extracted launch prices (your 105 drugs were launched at some point between 1981 and 2015) and then you took the launch price and adjusted it to 2015 price levels using the GDP deflator. You did not extract the 2015 price for all your drugs from the BNF (to look for example if the price increased beyond inflation). Is this correct?
---

	Which classification and which source did you use to classify drugs by drug class/mechanism of action and first of kind and follow-on drugs? Discussion p.17 “General pressures on pharmaceutical developers’ R&D costs do not explain the marked variations in the launch price of drugs across the five therapeutic areas that we observe in our study.” I do not understand this sentence: which general pressures on R&D costs? How is this conclusion drawn from your findings? You present price data but no R&D data (which as you say is hard to access and not transparent). p. 17 “the mean inflation-adjusted 28-day launch price for drugs indicated for colorectal cancer was 46 times greater than the price of those for hypertension”. I am a bit hesitant in comparing launch prices across therapeutic areas in a way that suggests there is no reason they should not be about the same level. I am not an expert in R&D but is it really fair to compare launch prices of colorectal cancer drugs with those of hypertension drugs? One should at least provide some context as to why they are expected to be similar or different. Limitation I think it would be have been very interesting to study price developments for each individual medicine over time rather than just launch prices. It would have been interesting to benchmark launch prices against some measure of therapeutic added value or cost-effectiveness. For future studies, you may want to consider using the rating by e.g. the French Drug Prescrire (seven levels) as done by other studies https://www.ncbi.nlm.nih.gov/pubmed/25967989 or, for solid tumours, the ESMO Magnitude of Clinical Benefit Scale. NICE Incremental Cost-Effectiveness Ratios would also be interesting to compare with.
--	---

VERSION 1 – AUTHOR RESPONSE

Reviewer: 1

Thank you for asking me to review this paper. The major increase in the numbers of new drugs and the relative escalation in costs over the past 10-20 years are of significant clinical and economic importance. Adding data and discussion to this debate is important and the authors have aimed to look at the wider fields of a number of medical specialties and allow comparisons in development and economics between them. I would support this aim but feel a number of changes to the paper could be made to improve the impact.

1/ Background information

It would be helpful to explain the choice of diseases used in the study. How many patients are on treatment for these conditions in the UK and how long does an average course of therapy last.

The disease were chosen on the basis of expert opinion to represent a range of physical and mental health disorders for which significant pharmaceutical innovation over the period of interest was known to have occurred. While all of these conditions are common, they were not chosen on the basis of the numbers of patients affected or their impact on health services (though we did aim to avoid rare or orphan conditions). We have updated the methods section to explain our selection in more detail.

2/ Study data

The title outlines that we are looking at New Drugs, however a number of the drugs in the analysis are old drugs with new indications (i.e. Methotrexate an old and cheap drug with an added indication for RA vs Rituximab a new and expensive drug with an added indication for RA). The costs of these drugs were already determined previously and are no different for the new indications. Should these drugs with new indications rather than new drugs be included in the study? The database may also include one error. 5-FU has been around a long time and I think may have been in the BNF since the 1970s. It would also be good to know the total number of new drugs for all indications licenced during the study period.

We believe that manufacturers frequently have a range of indications in mind when bringing new drugs to market and the order in which indications are licensed often represents progress in trial programmes and commercial factors; however, even when new indications for existing licensed drugs are more serendipitous, there is still merit in including them in our analysis as they still contribute to rising treatment costs for specific conditions. The majority of drugs included in our study were new drugs (new chemical entities or biologic agents), and our results section (and appendix) does highlight their role in price changes separately from new indications.

Thank you for identifying this error in our data; fluorouracil (5-FU) was initially licensed in the UK in the early 1970s. We have revisited the BNF entry that led to this error and it appears to have followed a marked difference in how the indications were reported, leading us to conclude wrongly, that this was a new indication for an existing licensed drug. We have updated our analyses, as well as the results section, figures and appendix, to reflect the removal of 5-FU from our dataset.

We have amended the results section to include a reference to show the proportion of all new drugs that our sample represents for the period of interest.

3/ Cost for 28 days

Whilst this is important, I am not sure if this is the only important comparison for economics. Some of the conditions (hypertension) are associated with very long treatment periods (decades) whilst colorectal cancer is usually only treated for a number of months. Some comment on the overall cost of treatment with a new therapy would also be informative. I may need to recheck my data but I think the BNF price for a 28 day cycle (21 day treatment) of Regorafenib was £3744 at launch but is £1296 in the data set.

We chose to use the price for a 28-day supply in order to allow comparisons against a common metric. We acknowledge that our decision to standardise comparisons in this way could be criticised as failing to note that treatment for some conditions and with some medications could potentially be lifelong, and hence generate much higher total costs for payers. Indeed, some innovations in cancer could be life-extending, and so overall costs could escalate further, albeit with important health gains. However, our principal comparisons were within disease areas, and in general, this means treatments are given for a similar duration or course.

Thank you also for identifying this additional error in our data; the price for regorafenib at launch is as you stated. The results section, figures and appendix have been updated to show the correct price.

Additional info

To help get a wider of view of this, it might be helpful to consider adding some more detail to the discussion. The authors describe the costs of developing a new drug and this impact but do not supply an estimate of this cost. This would aid the discussion. It may be worth adding in some discussion on how UK drug prices are set. The UK has a number of price control mechanisms PPRS and NICE and some readers may be unaware of the very significant impact these have on drug

prices. Overall this comparison showing the impact of new technology and new drugs on the economics of differing types of illness is welcome and should be of value to those looking at the overall health economy, rather than only from an individual specialist viewpoint

There is no single accepted estimate for the costs of developing a new drug through all its stages, and no estimate of the different costs of developing new drugs in different therapy areas, so we have not attempted to supply this. We have amended the discussion to remove the sentence on pharmaceutical developers' R&D costs and expanded the subsequent sentences to make clear that we cannot account for the different drug development costs in different therapy areas.

We amended the introduction to include the PPRS scheme in the introduction and mention NICE in the discussion section (as the specific and relevant HTA body for England and Wales).

Reviewer: 2

The title indicates a much larger study. It does not indicate that only five health conditions were focused. Many readers, looking at the heading, may be excited, but disappointed when reading the paper. The title should be rewritten.

We have amended the title as suggested.

The major weakness of this work is that the price paid by the National Health Service (NHS) is not known. Today negotiation over price is common in most countries and secret pricing is a huge problem when health economic analyses are undertaken. The price actually paid may be less than half of the official list price. I strongly recommend that this weakness is more elucidated and the price level actually paid should be given or at least indicated. The secret pricing and the fact that the results of the negotiations over price are forbidden to publish, causes a significant problem to health administrators and economists struggling with the documentation of cost effectiveness in Health Technology Analyses (HTAs). The great challenge is not the rising costs/prices, but the fact that almost no-one pay these prices.

As the reviewer notes, the actual price paid by local or national NHS providers is not actually known; it remains commercial in confidence, so we cannot determine this. For a service-wide perspective, it would be possible to report on the total medicines budget, but this would not provide information on the proportion spent on new drugs and would not allow an exploration of the price paid for drugs indicated for a specific condition. Instead, we must rely on the published list price for drugs. We acknowledge that this may not be actual price paid by NHS organisations, particularly for more recently launched drugs, as this does not account for the effects of local price negotiations and the increasing use of patient access schemes, which have become more common. Such schemes predominate in new cancer drugs, and tend to be of an order sufficient to yield a cost-utility ratio acceptable to NICE, but their impact would tend to blunt the apparent increase in price over time for a particular condition, whereas our data still show a marked increase in the inflation-adjusted price of new drugs for colorectal cancer. We have amended and expanded the strengths and weaknesses section of our discussion to account for this limitation in our data.

The 28 days perspective is very short. Some therapies are administered for a limited time frame (in ex. colorectal cancer therapy) and others are administered for decades (hypertension, arthritis etc.). Despite a low cost per month, the cost to the society may be high. This should also be more strongly pointed at.

Please see our response to the same issue raised by Reviewer 1.

Reviewer: 3

This is an interesting and well executed study. I think the study could have added more value if it had collected also some indicators of the drug added clinical value to put prices into perspective and if it has evaluated price development for the individual drugs over time. So overall the study provides interesting longitudinal evidence on launch prices and raises important questions as to price drivers, it does less in terms of providing answers in regards to determinants launch prices over time (not the objective of the study but an important question to answer from a policy and practice perspective).

We agree that both the clinical value of new drugs and the actual determinants of launch prices are important considerations, but they are beyond the scope of this paper. The former is regularly discussed in our team, who are close observers of NICE. The latter would really require “fly on the wall” observations within pharmaceutical developers’ internal marketing meetings. Both are hinted at in our discussion section, which we have also amended to make the scope of our study clearer.

Background

p.5 Can you add the relevant references to the last two sentences:

“In the USA, [.]”

“According to a recent [..]”

We have included the relevant reference as requested.

It is important to distinguish clearly between price and expenditure. The focus of the article is on the price, price can be a driver or increased expenditure but expenditure could also increase independently of a price increase (move towards newer and more expensive products). I think it would be good to clearly distinguish price from expenditure as well as their respective drivers. I would perhaps avoid introducing a third term ‘cost’ to avoid confusion. In the second paragraph you talk about affordability. Affordability is driven by price and volume however the rest of the paragraph focuses on price only. There are other tools available to health care systems to ensure access (of which affordability is a major component as you say). Use of limited formularies (essential medicine lists) is a tool to limit use to the most cost-effective medicines which contributes to better affordability for health care system for example. I suggest to improve the flow of the paragraphs leading up to the objective of the study: expenditure on medicines is increasing, there are several drivers, price is one of them, your article focuses on prices of new medicines (you may also want to acknowledge that there have been price increases for off-patent medicines but that this is not the focus of your study).

Thank you for your very helpful suggestions. We have amended the background and introductory paragraphs to distinguish these concepts and improve the flow.

It may not always be possible, but where these are available, I would suggest to use more recent references in your background (particularly when referencing papers on expenditure and price trends). Here some examples of recent articles you may want to consider citing:

<http://care.diabetesjournals.org/content/early/2018/05/03/dci18-0019>

<https://www.ncbi.nlm.nih.gov/pubmed/29016226>

<https://www.ncbi.nlm.nih.gov/pubmed/28605615>

Thank you for these interesting additional references. The latter two deal with trends in price post-launch; while relevant for broader discussions of pharmaceutical price inflation, this is not the issue dealt with by our current study, and in any case post-marketing prices are regulated by the PPRS scheme in the UK (now mentioned in our Background/introductory section).

Methods

Overall, methods are clear and well described.

p.8 'launch price for a typical 28-day supply and 2015 adjusted 28-day launch price were extracted'

I am not sure if I understand this correctly: You extracted launch prices (your 105 drugs were launched at some point between 1981 and 2015) and then you took the launch price and adjusted it to 2015 price levels using the GDP deflator. You did not extract the 2015 price for all your drugs from the BNF (to look for example if the price increased beyond inflation). Is this correct?

Yes, this is correct. We used actual prices from the year of launch and inflated these to 2015 prices.

Which classification and which source did you use to classify drugs by drug class/mechanism of action and first of kind and follow-on drugs?

Where the mechanism of action or drug class was not stated in the BNF, it was obtained from the commercial Pharmaprojects database. We have amended the methods section to describe this.

Discussion

p.17 "General pressures on pharmaceutical developers' R&D costs do not explain the marked variations in the launch price of drugs across the five therapeutic areas that we observe in our study." I do not understand this sentence: which general pressures on R&D costs? How is this conclusion drawn from your findings? You present price data but no R&D data (which as you say is hard to access and not transparent).

We agree, with hindsight, that this sentence is confusing and so have deleted it from the discussion, though developers themselves do emphasise research cost escalation as a key driver of increasing drug prices. We have expanded the subsequent sentences in the discussion to make clear that we cannot account for the different drug development costs in different therapy areas. Please also see our response to Reviewer 1.

p. 17 "the mean inflation-adjusted 28-day launch price for drugs indicated for colorectal cancer was 46 times greater than the price of those for hypertension". I am a bit hesitant in comparing launch prices across therapeutic areas in a way that suggests there is no reason they should not be about the same level. I am not an expert in R&D but is it really fair to compare launch prices of colorectal cancer drugs with those of hypertension drugs? One should at least provide some context as to why they are expected to be similar or different.

We have expanded our discussion section, in particular the strengths and limitations element, to describe the rationale for this approach in more detail, and acknowledge the potential criticisms that may arise. Please also see our response to Reviewer 1.

Limitation

I think it would have been very interesting to study price developments for each individual medicine over time rather than just launch prices. It would have been interesting to benchmark launch prices against some measure of therapeutic added value or cost-effectiveness. For future studies, you may want to consider using the rating by e.g. the French Drug Prescrire (seven levels) as done by other studies <https://www.ncbi.nlm.nih.gov/pubmed/25967989> or, for solid tumours, the ESMO Magnitude of Clinical Benefit Scale. NICE Incremental Cost-Effectiveness Ratios would also be interesting to compare with.

Thank you for your suggestions for further studies, in particular the suggested sources of information on clinical benefit. We will most definitely take these on board within our team when considering further work in this area.

VERSION 2 – REVIEW

REVIEWER	Dr Philip Savage BSUH, Brighton, UK
REVIEW RETURNED	19-Feb-2019

GENERAL COMMENTS	This paper addresses issues of drug price escalation by comparing across differing types of diagnosis. The paper has been revised to address the earlier issues and is now fine for publication from the oncology view point, colorectal cancer is one of the diagnoses with the least impact of new drug technology over the study period. I suspect one of the major other tumor types would have shown much more new drugs and info on pricing. However economic comparisons between disease types are interesting for bot the economist and the clinician.
--

REVIEWER	Alessandra Ferrario Postdoctoral Research Fellow Division of Health Policy and Insurance Research Department of Population Medicine Harvard Medical School and Harvard Pilgrim Health Care Institute Boston, USA
REVIEW RETURNED	25-Feb-2019

GENERAL COMMENTS	Thank you for addressing the comments raised.
---